# A framework for promoting online prosocial behavior via digital interventions

David J. Grüning [1,2,3 ✉], Julia Kamin[3,4], Folco Panizza[3,5],
Matthew Katsaros [3,6] & Philipp Lorenz-Spreen [3,7]

Digital interventions for prosocial behavior are increasingly being studied by psychologists. However, academic findings remain largely underutilized by practitioners. We present a practical review and framework for distinguishing three categories of digital interventions--proactive, interactive, and reactive--based on the timing of their implementation. For each category, we present digital, scalable, automated, and scientifically tested interventions and review their empirical evidence. We provide tips for applying these interventions and advice for successful collaborations between academic researchers and practitioners.

D igital interventions are one of the primary tools for promoting prosocial and reducing antisocial behavior online. We define digital interventions for prosocial behavior as any type of modification in the design of a platform, related to its architecture or rules, that aims to promote prosocial or minimize antisocial behavior in the interactions between the platform's users. We acknowledge that there are digital interventions that are not specifically implemented in a digital platform, such as smartphone applications or browser extensions. However, platform interventions are the focus of this paper due to their undisputed prominence in intervention research and practice. There is an abundance of digital interventions that aim to increase prosocial behavior online through various psychological techniques such as explicit transparency[1], inoculation[2], and mental friction[3,4]. The field of digital interventions for prosocial behavior is growing rapidly[5–9] and addressing deeply entrenched social problems such as radicalization[10], bullying[11], and misinformation[12] on online platforms.

While the growing body of research on digital interventions speaks to scholars' awareness of these social problems in the digital space and their interest in moderators of online interaction, it remains unclear how readily these findings are used by practitioners such as UX designers and product managers. Do practitioners turn to research published in academic journals to learn about viable digital interventions when building and designing online platforms? Based on anecdotal data, we do not think that's the case. We sought to answer this question in a qualitative survey of 35 practitioners in the field of social media and online design (all data is accessible in an open repository, OSF: https://osf.io/b3es6/?view_only=e5aa0dbeb16c4781986f340c38f89482). The participants held different roles--from UX designer to product manager to software engineer and developer - and had different levels of expertise, ranging from 1 to over 13 years of experience in the field. All respondents agreed that ethical and social approaches to the design of online exchanges is an important facet of social media (80% said it was most important), and many said their company or organization considered it important (31%) or very important (54%) overall. Significantly, the overwhelming majority (86%) said that in their organization, evidence is an important or very important requirement for new design decisions such as implementing digital interventions. However, only a minority (37%) reported regularly

[1] Heidelberg University, Heidelberg, Germany. [2] GESIS – Leibniz Institute for the Social Sciences, Mannheim, Germany. [3] Prosocial Design Network, New York, NY, USA. [4] Civic Health Project, New York, NY, USA. [5] IMT School for Advanced Studies Lucca, Lucca, Italy. [6] Yale Law School, New Haven, CT, USA. [7] Max-Planck Institute for Human Development, Center for Adaptive Rationality, Berlin, Germany. ✉email: david@prosocialdesign.org

consulting evidence from academic research; instead, all practitioners surveyed reported relying on testing by other companies or within their own organization We suggest that this under-utilization of scientific research is the result of the inaccessibility of scientific knowledge due to economic (e.g., journal subscriptions) and communication barriers (e.g., scientific and technical jargon), the time constraints of reading individual scientific papers, the large number of imprints in which results are published, and the focus of studies on experimental rigor rather than practical utility.

In this paper, we attempt to present a framework of existing digital interventions that promote prosocial interactions online. We categorize interventions into three types based on their timing with respect to when a relevant behavior occurs. For each category, we provide several examples of scientifically tested interventions in the interest of furthering the scientific study of interventions and also encouraging their application by designers and others in the digital community. We provide a critical review of the existing research for each intervention, including its strengths, weaknesses, and areas for future research. We conclude each of the three intervention categories with a summary and outlook for their respective interventions. Importantly, two additional steps in the paper make the framework particularly applicable for practitioners. Specifically, we comment on how to apply interventions from different framework categories to real-world platform problems. Finally, we provide an overview of existing formats for collaboration between practitioners and researchers to test digital interventions for prosociality on digital platforms.

## A framework of digital interventions for online prosociality

We use four criteria to categorize interventions as applicable to large-scale online settings (Fig. 1).

First, interventions must be digital. Purely analog interventions (e.g., meditation techniques) and interventions that do not directly affect the digital environment (e.g., listening to soothing music) were not included in our framework. Analogous tools for reducing online prosocial behavior may be viable but are not specifically relevant to practitioners on digital platforms.

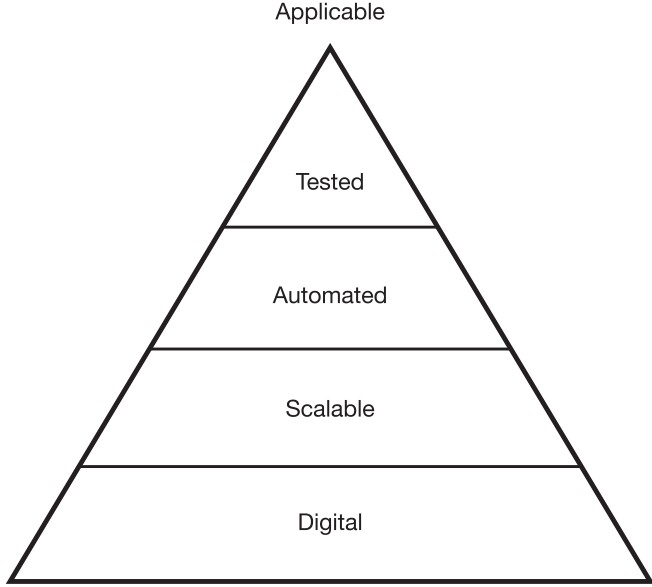

**Fig. 1 Pyramid of consecutive criteria to be met for applicable interventions.** Pyramid visualising the four subsequent criteria for a digital intervention to be applicable.

Second, interventions must be scalable. Scalability refers to the realistic potential for an intervention to be applied to a large number of users, either directly or indirectly through network effects. We focus on scalable interventions because they have the greatest potential for impact.

Third, digital and scalable interventions must be automated. This requirement is closely related to our scalability criterion, but we believe it applies to a smaller set of interventions. While all automated interventions should be scalable, not all scalable interventions need to be automated. For example, Stroud et al.[13]. showed that having a journalist participate in a social media discussion had a positive effect on the nature of the discussion and the general climate of opinion. Bringing such experts into the dialogue (such as professional fact-checkers and regulators of new posts and comments hired by Facebook and Twitter, now known as X) may be scalable, but only with a large investment of material and human resources. Automated interventions may be scalable with relatively few resources.

Fourth, we only include digital interventions that have been empirically tested. Interventions were considered tested if at least one controlled or natural experiment provided evidence for their proposed effect(s). This meant that the evidence for an intervention was published (i.e., peer-reviewed or preprinted) and allowed for independent verification of methods, data, and analyses.

Applicable interventions that met all four criteria were assigned to one of the three proposed categories of digital interventions for prosocial behavior.

**Categorizing interventions by deployment time**. We propose that research on digital interventions can be divided into three categories that might help practitioners understand which intervention is more appropriate for their case: proactive, interactive, and reactive interventions. The categorization is based on the general idea of temporal starting points of interventions in non-digital settings, such as proactive interventions (e.g., alcohol abuse[14]; smoking[15,16]). The point in time at which the intervention occurs significantly alters its effects (e.g., from proactively reducing risk or interactively influencing content to a more prosocial direction) and thus determines its use cases. This categorization applies to digital interventions regardless of an intervention's intended outcome––such as reducing polarization or promoting supportive engagement. Importantly, the categorization is practically actionable, i.e., it is informative about how to implement interventions of a certain type in the digital space to address a particular problem (e.g., stopping further polarization in its tracks).

The three categories relate the timing of the intervention to the behaviors it is designed to address. Proactive interventions aim to promote prosocial behavior or stop antisocial behavior before it occurs. Other interventions are interactive, that is, they operate while the targeted problem is occurring. The final group of interventions is reactive: this type of intervention follows the behavior in question, reducing or increasing the likelihood that it will occur in the future. We provide examples of interventions for each category.

In the following, we outline examples of proactive, interactive, and reactive digital interventions for prosocial behavior: Each intervention is described in terms of what it is and when it should be used. We also outline the intended impact and the scientific evidence behind each intervention (additional information about the interventions is available in the library of the Prosocial Design Network; https://www.prosocialdesign.org/). Table 1 lists all interventions, their category, intended impact, and supporting empirical evidence. We also provide examples of practical

**Table 1 List of interventions, their category, intended impact, empirical evidence, and illustrative real-world implementations.**

| Intervention | Type | Intended impact | Evidence | Illustrative Implementation |
|---|---|---|---|---|
| Reminder of norms | Proactive & Reactive | Increase adherence to platform rules among new users and rule-breakers | Matias (2019)[1,11]; | Twitch "Chat Rules" (extensive presentation, Toner, 2017) |
| Prebunking | Proactive | Inoculate users against misinformation | 2,56 | YouTube Hit Pause interstitials; Online games[1-4,57-60] |
| Tips for headline checking | Proactive | Reduce spread of mis- and disinformation | 22,61,62 | Google[1-3]; Facebook; Instagram |
| Prompt about trolls | Proactive | Make new users more comfortable in community and more resilient to trolling | 27 | |
| Prompt to rate accuracy | Interactive | Reduce spread of mis- and disinformation | 28,29 | Prompts appearing right after participants decide to engage with content |
| Labeling misleading content | Interactive | Reduce spread of true but deliberately misleading information | 12,34-36,63 | Twitter Community Notes |
| Preliminary flagging | Interactive | Reduce online harassment | 38,39,42 | Instagram[1,2]; Nextdoor[1,2]; Pinterest; TikTok; Twitter, now known as X; YouTube |
| "Thank you"-button | Reactive | Increase retention & contagious prosocial behaviors | 18 | Bumble |
| Removal explanation | Reactive | Reduce repeat rule-breaking | 42,43 | |

implementations. Notably, the majority of digital interventions do not directly promote prosocial behavior, but rather attempt to reduce antisocial behavior (but see other examples[17,18]), an imbalance that is reflected in the examples included in this paper.

## Proactive Interventions

Proactive interventions aim to promote prosocial behavior or stop antisocial behavior in its tracks by reducing the likelihood that problematic behavior will occur.

**Reminder of norms**. Reminders of norms are designed to remind users of the social norms of a forum. There are two versions of this intervention, which differ in their targeting.

The reminder intervention is used before users comment on an online forum. It includes a message to new users at the top of the joint forum or platform that includes a welcome note, a reminder of the community rules (with a link to the full set of rules), and a mention of how and by whom these rules are enforced (e.g., moderators and community members). This message is visible to all new users who join the community.

With this intervention, users should be more likely to follow the stated norms of the platform or community. New users who see the message are more likely to engage with the community in general because they are confident in their ability to contribute thoughts and opinions that adhere to the norms of the group and platform, and because they are confident that their comments will not result in abuse. Matias et al.[11] conducted a large-scale field experiment in the Reddit community r/science, in which posts were randomly assigned to receive a norm reminder. Posts that received the intervention were 70% more likely to have new users comment on the thread, and 8% more likely to have comments from new users that followed community rules. Notably, the community rules for the r/science community under study were unusually strict (e.g., no personal anecdotes allowed in discussions), and newcomer deviation from these rules had previously been unusually high. This presumably made the reminder intervention more effective than in other communities. However, Kim et al.[17] demonstrated a similar effect on the networking platform Nextdoor: pointing out prosocial community guidelines

before joining a group reduced the number of reports on the platform and improved the moral grounding of comments.

Although norm reminders are an effective proactive intervention, they can also be used reactively as a targeted intervention after users have had their content removed for violating community rules. In this case, a message would be posted at the top of the comment forum or platform, visible only to the offending user. As with the proactive intervention, the message reiterates the community rules.

The post-removal intervention is intended to increase rule-breakers' compliance with the rules for subsequent posts they contribute to the platform. Indeed, in a large-scale field experiment on Facebook, Tyler et al.[1] showed that a norm reminder at the top of a user's feed increased rule-breakers' adherence over the next 45 days. Specifically, the post-reminder intervention reduced both repeat rule-breaking and appeals of removals. In addition, the rate at which appeals were successfully overturned increased, and users perceived the removal process as more transparent and fair.

The empirical evidence for the effectiveness of norm reminders in increasing compliance is strong and readily applicable to communities with specific rules and norms. However, on platforms with less transparent rules and heterogeneous norms, its application seems less straightforward, and thus its scalability needs to be further improved and tested for such platforms. In this context, it's worth noting that a replication study[19] yielded mixed results. Specifically, the authors repeated a similar study on r/science and two additional Reddit communities, finding a smaller but consistent effect in the former, but no effect in the two newly tested communities. In addition to differences in the relative volume of new users entering the communities, the authors suggested that norm reminders may be particularly effective in communities where there are well-defined and widely communicated rules.

**Prebunking**. A prebunking intervention consists of showing users messages via pop-ups that inoculate them against misinformation. These messages either inform users about flawed arguments commonly used in misinformation or highlight the scientific consensus regarding potential future misinformation.

The messages can be set up to appear as unsolicited prompts or in a more interactive format when users search for terms or hashtags related to artificially contested or polarized issues (e.g., global warming, autism, or vaccines) or anticipated narratives (e.g., election fraud).

Prebunking interventions aim to reduce the number of users who interact with and subsequently spread misinformation about an issue, and to reduce the extent to which users believe the misinformation presented. Without judging the content, this intervention introduces scientifically based facts into the discussion before the potential encounter with misinformation. This intervention also identifies common argumentation techniques (independent of a specific misinformation topic), thereby making social media audiences more resilient to these techniques in general. According to Lewandowsky and van der Linden[2], the most common argumentative techniques that readers can look for as predictors of misinformation are incoherence (e.g., "Global temperatures can't be measured accurately, but we shouldn't worry because they've been cooling for the past 5 years"[20]), false dichotomies (e.g., "Either you are with us or you are with the terrorists"[20]), scapegoating, ad hominem attacks disguised as arguments, and emotional manipulation. Lewandowsky and van der Linden[2] further explored proactive measures to prevent misinformation from gaining traction, based on the psychological theory of "inoculation. Inoculation is based on the idea that if people are warned that they might be misinformed and exposed to examples of how they might be misled, they can learn to resist and become more immune to misinformation over time. The researchers reviewed diverse techniques for increasing people's resilience to misinformation. Their review concluded that vaccination interventions of different types can be effective and are a promising way to protect people from misinformation and fake news. A reduction in the spread of misinformation has been attributed to increased discrimination between true and false news; however, recent analyses suggest that prebunking may simply make users more conservative in their judgments, namely, more skeptical of news content in general (see, e.g., for inoculation interventions[21]).

Prebunking appears to be a promising set of interventions because a growing body of empirical evidence supports their effectiveness, they are not content-specific, and they can be implemented without actively tracking the problematic behavior (i.e., sharing misinformation). They are well-tested and scalable. However, the potential spillover effects, such as a general loss of trust in the media[21], should be tested in future studies and monitored during any practical implementation on a platform.

**Providing tips for headline checking**. This intervention is based on providing users with a list of easy-to-use headline-checking tips. For example, tips may serve a general priming function (e.g., "Be skeptical of headlines"), reveal a particular persuasion technique (e.g., "False news stories often have catchy headlines in all caps with exclamation points"), and teach a specific decision rule (e.g., "If shocking claims in the headline sound unbelievable, they probably are"). Such tips or a list of tips should appear either as an intermittent reminder or as a public service announcement. They can even be implemented as an interactive intervention, alerting users to consider the accuracy of headlines before sharing news articles.

Presenting tips to users is intended to make them significantly more skeptical of false news and, as a result, less likely to spread misinformation. Guess et al.[22]. exposed participants in both the United States and India to a media literacy intervention. In the study, participants were presented with reminders about how to judge the accuracy of a news source. The authors found a

significant increase in skepticism about the veracity of the articles and (hyperpartisan) headlines they read, and a decreased willingness to share such news. Other findings also support the effectiveness of this intervention[23–25] and the impact of related interventions such as lateral reading[26].

Although the evidence is still sparse, the provision of online content control techniques seems promising because they have been tested in different cultural contexts, are content-independent, and can be easily applied to a wide range of users, making them partially empirically supported and scalable. However, the specific tips and strategies are potentially subject to change (as misinformation techniques change) and need to be kept up to date in order to be an actionable intervention. In particular, techniques such as lateral reading that go beyond media literacy adapted from offline media need more testing in the future, for example, to study their impact over time.

**Prompt about trolls**. This intervention is a message to new users of a platform or online community that has three key elements. First, the message includes a link that describes what is considered trolling in the community. Second, it warns users that they will observe trolling from time to time, but that they can rest assured that the good-faith users outnumber these provocateurs. Finally, the message provides instructions for dealing with trolls. The prompt can appear as an automated comment on a new user's post, as an automated personal message, or as an interstitial upon entering the community.

This intervention is intended to increase the retention of new users who have joined the community in good faith. With the notification, users know in advance that trolls are present and have the tools to deal with them. As a result, they are less likely to be driven off the platform by harassment. The intervention may also increase the proportion of good-faith users relative to trolls. Matias et al.[27]. conducted a two-year project with the r/feminism community on Reddit to test the prompt intervention to reduce harassment. After presenting this intervention to 1300 users, they found that messages explaining that harassers were a minority increased newcomer comments by 20% on average. This effect persisted over the entire 10-week period of the intervention.

It's worth noting that the evidence for this intervention is limited to a single study, but one that represents a long-term field approach. Given that the study was conducted in a single community (r/feminism) on a single platform (Reddit), more work is needed to demonstrate the generalizability and potential impact of prompting about trolls. The scalability of the intervention, and the fact that it can be automated with little effort, make research on the intervention in different online forums and communities even more urgent.

## Conclusion

Proactive interventions are the most researched form of intervention in digital environments. This may be due to their idealistic format of independence from the antisocial behaviors they target and, in an optimal case, their ability to stop these behaviors in their tracks. While some proactive interventions have been rigorously validated and can now be tested or relied upon in applied contexts (i.e., reminder of norms and prebunking), some newly conceptualized interventions require more fundamental testing (i.e., tips for headline checking and prompt about trolls).

## Interactive interventions

Interactive interventions occur at the exact moment the target behavior occurs. They are designed to mitigate negative behavior or reduce its impact.

**Prompting attention to accuracy**. This prompt intervention alerts users to the accuracy of new news headlines they encounter. It appears as an interstitial before a user shares a post or link, while they are still considering their post intent and any content they might add.

Users exposed to this prompt should be more likely to share higher-quality news and sources and be more selective about the content they interact with on platforms and in communities (e.g., likes and comments). Companies may be concerned that prompts will reduce overall engagement on their platforms; however, research by Capraro and Celadin[28] suggests that when prompts are presented alongside a share button, the interactive intervention significantly reduces engagement only with false or misleading content. Such a prompt can also consist of periodically asking users to rate the accuracy of content they observe online[29,30].

The effect sizes of accuracy nudges are small but have been replicated in multiple studies. Their implementation is scalable, as they are a low-effort intervention. However, it is unclear how accuracy motives interact with other motives for sharing information online[31,32] and with political partisanship[33]. Generalizability to other forms of prompting remains largely untested.

**Labelling misleading content**. This intervention refers to attaching informative nudging labels (e.g., "Be skeptical!" or "Misleading") to posts that are not technically false but are easily and commonly misinterpreted. For example, misleading posts would be labeled as such and links added to reliable and up-to-date information on the topic, or logical fallacies in a particular post would be highlighted. The intervention is suitable for any social media area that relies on accurate information and its unbiased processing. The intervention message would appear as people view content that has been shared.

Marking misleading content is intended to reduce the spread of misinformation in the form of misrepresented or intentionally distorted facts. Indeed, Yaqub et al.[34]. found that labeling online content as controversial reduced the likelihood that readers would share the labeled content. Moreover, the labeling effect appears to be independent of political views[35], but dependent on the actual news label (e.g., "disputed" performs worse than "rated false"). Selective labeling of certain posts over others could also lead to the implied truth effect: Misleading information that is not explicitly labeled, as opposed to labeled information, may be interpreted as more likely to be true[12]. Moreover, presenting the label immediately after the content rather than immediately before or at the same time seems to increase its effectiveness, possibly because reading the content itself enhances the effect of the label[36,37]. The intervention of labeling content as misleading has a convincing amount of empirical evidence. However, the practical implementation of this intervention requires extraordinary resources and an automated decision mechanism to accurately label content that is truly misleading as such.

**Preliminary flagging**. This is an AI-based intervention. First, when a user attempts to post a comment, an algorithm is used to assess the toxicity of the comment's language. Comments that exceed a preset toxicity threshold then trigger a prompt that asks the user if they want to reconsider or revise their post. This can be used in conjunction with any comment section or platform that relies primarily on short text posts. The suggestion should appear as an interstitial to the user after they have pressed the submit button, but critically, before the post goes live for other users. The user should be presented with options to edit their comment, cancel the comment, or post it as is.

The main goal of this intervention is to reduce the amount of online harassment perpetuated by toxic comments. In particular, it aims to help people slow down and reconsider their comments and to avoid posting something they might later regret. A study by Simon[38] on their own platform found that about half of users either revise their comment or decide not to post it when prompted that their comment might be inflammatory. In addition, in a randomized controlled experiment conducted on Twitter, now known as X, Katsaros et al.[39]. found that users prompted with this intervention posted 6% fewer offensive tweets than non-prompted users. According to the researchers, this decrease in the creation of offensive content could be attributed not only to increased deletion and revision of prompted tweets, but also to a decrease in the creation of future offensive tweets and replies. There is a growing body of evidence that suggests promising directions for these preliminary interventions against toxic language. While not content-neutral, this is a scalable intervention through automated analysis of written text, while ultimately leaving the agency with the users. The potential collective effects on the network propagation of toxic language are of particular interest for further research and application. Furthermore, research in other contexts (e.g., prompting transparency and fairness in regards to procedural justice theory[40,41]) would suggest that the way the short message in the prompt is framed would have an impact on the likelihood of people deleting their comment. Future research can explore how different messages can be used in this intervention to further increase the impact of the intervention.

**Conclusion**. The literature on digital interventions contains a considerable amount of research on interactive interventions. It seems particularly important to note that the most extensively researched interactive interventions are also the subject of extensive debate about their effectiveness. That is, for both the accuracy nudge and the misleading content labeling, the evidence is mixed and suggests that the effectiveness of the interventions varies substantially depending on additional factors (e.g., intervention content[35]; and user intentions and motives[31,32]). In summary, the most urgent future research direction for interactive interventions is the collective and unbiased approach of synthesizing existing results and deriving robust tests of specific questions that remain unanswered (e.g., the moderation effect of the content of intervention messages and short- vs. long-term effectiveness of interventions). Meta-analyses and large-scale collaboration projects are in need to identify and test, respectively, the boundary conditions of interactive interventions.

### Reactive interventions
Reactive interventions follow the behavior in question, reducing or increasing the likelihood of it recurring in the future.

**Thank-you button**. This intervention is an interactive button or reaction icon that allows users to send positive signals to others. Unlike the Like-button, it allows users to specifically express their thanks and gratitude to someone for their contribution to an online community. The button works either as an additional react icon for when users not only like the content but also appreciate the creator, or as a standalone button below the content.

The Thank-you button is intended to encourage more prosocial behavior by providing a new and more visible way to thank and ultimately reward people for high-quality contributions. Matias et al.[18]. studied the effects of 344 volunteers thanking 2,702 Arabic, German, Polish, and Persian language Wikipedia contributors over the course of a year. The authors showed that the use of a thank you button increased the amount

of interactive prosocial signaling, with more users sharing their gratitude with others. It also boosted the short- to medium-term retention rate of Wikipedia contributors.

Although the evidence for the thank-you button intervention is sparse, and Matias' study was conducted in the context of an open collaborative project (i.e., Wikipedia) rather than a social media platform, the conceptual idea has obvious potential. Allowing users to send additional positive signals is highly scalable, as it is applied by the users themselves. This also makes the "thank you" button intervention a likely candidate for powerful network effects of positive signaling. Given the sparse scientific evidence, but potential and testing in limited contexts, we encourage practitioners and researchers to explore this intervention together.

**Removal explanation**. In the event of a rule violation and the subsequent removal of the violating content, this intervention provides the offending user with an explanation of why their content (e.g., post or comment) was removed. After receiving a comprehensible explanation, users are less likely to violate a forum's rules in their future contents or appeal the removal.

In an observational study, Jhaver et al.[42]. showed that Reddit users were less likely to break another rule when they were given an explanation for why their content was removed. This effect was stronger when detailed explanations, as opposed to short sentences, were provided. Encouragingly, the authors did not find a difference in subsequent rule-following when the explanations were provided by a real human compared to messages provided by a bot. Srinivasan et al.[43]. replicated this finding in a natural experiment. Notably, however, the authors did not find a reduction in toxic content per se, but rather that users were more likely to avoid certain wording and phrases that violated the platform's rules. That is, users seemed to be more aware of the community's specific rules, rather than being discouraged from engaging in toxic behavior.

This latter finding points to a potential loophole in the removal explanation intervention: clearly communicated community standards can be actively circumvented, and more robust rules may be needed for this intervention to be successful in the future.

**Conclusion**. The reactive intervention format is the least researched of the digital interventions. There are many concepts for useful and applicable reactive interventions in the digital space (e.g., thank you button) that have either been proposed or tested by platforms without providing detailed descriptions of the study designs and without sharing their data. In particular, some of the most widely used interventions on social platforms are reactive––such as content removal, content filtering, and user removal. These interventions were not included in this framework due to the limited empirical literature on them. The category of reactive interventions has much room to develop its practical concepts into scientifically tested digital interventions. We strongly encourage more scientific study of these basic interventions to understand the impact of these already widely used tools.

## Applying interventions

The framework presented is intended to provide practitioners with insights into specific interventions that can be used to promote prosociality on their platforms. The goal of this work is not to suggest that every platform should adopt all these interventions, but rather to serve as a way to collect, organize, and present the available scientific evidence to practitioners in a way that can be useful for their work building our collective social online spaces.

One way we envision practitioners using this framework is to evaluate their current set of interventions and products. While we do not suggest a perfect mix of proactive, interactive, and reactive interventions that a platform should offer, we hope that platforms will consider building interventions that address problems at each of these stages.

In addition, this framework and the interventions provided can be useful as a tool for individual practitioners to brainstorm and design new interventions using scientific evidence as inspiration. Particularly when paired with commonly used design tools such as user journey mapping[44], the framework presented can help teams think through the point - proactive, interactive, or reactive - at which intervention might be most appropriate and suggest contextually appropriate solutions.

Finally, this framework can be helpful in extending and repurposing existing interventions. Some of the interventions presented above may fall into different categories depending on how they are applied. For example, norm reminders can be effective when presented to someone who's joining a new space (proactive) or immediately after someone breaks a rule (reactive). Practitioners who have existing interventions that their users are familiar with can use this framework to brainstorm how to take an existing reactive intervention and make it proactive.

## Collaboration between practitioners and researchers

There are a variety of approaches to practitioner-researcher collaboration. Each approach has its own strengths and weaknesses, and the choice of method depends on specific goals, ethical considerations, and the regulatory environment.

One avenue for collaboration is for platforms to share their data with researchers, allowing them to conduct studies with access to real-world user information. This may include collaboration on the planned study itself. The approach promotes active transparency on the part of the platforms and allows for independent and therefore more accurate results on the part of the researchers. However, this format can raise concerns about user privacy, and the scope of the research may need to be limited by the platform's data-sharing policies. Past examples of this format have been organized at Twitter, now known as X (https://developers.tiktok.com/products/research-api/), TikTok (https://developers.tiktok.com/products/research-api/), and as systematic approaches from the academic side (e.g., Harvard's Social Science One; https://socialscience.one/).

Another prominent approach is for researchers to access data directly (e.g., using APIs) without explicit collaboration with the platform. This can lead to unexpected discoveries, but it raises ethical and legal questions from the platform's perspective (due to the risk of unauthorized data access) and leaves uncertainties about control over what user information is analyzed in what detail. A working format for such independent research could be an agreed guideline (see e.g. https://independenttechresearch.org/).

Community-driven research involves the active participation of users (e.g., https://citizensandtech.org/), either independent of the platform itself or organized in accordance with the platform and the researchers. We want to emphasize this form of collaboration because it promotes the co-creation of solutions by the actual user base. As a potential drawback, such collaboration with up to three different parties requires careful management.

Finally, platform-internal research projects can ensure a close working relationship between practitioners and researchers, with the possibility of experimental testing that is unique to this type of collaboration[45]. However, there is also the risk of a conflict of interest between what the company wants to communicate to its public audience, especially its direct users, and what the project's results show. This conflict is further complicated by the

dependence of researchers on the platform[45]. This hierarchy can lead to a lack of transparency and limited invitations for the external scrutiny that seems to be the backbone of scientific endeavor (i.e., cross-validation by independent evaluators).

To collaborate effectively, practitioners and researchers must weigh the pros and cons of these different approaches, taking into account their specific intervention goals, ethical considerations of the platform and the research being conducted, and the regulatory environment from a platform and broader policy perspective. Regardless of individual efforts to collaborate between platforms and researchers, overarching regulations and transparent guidelines for these regulations must be a high priority. Efforts such as the Digital Services Act (DSA), while imperfect, are a step in the right direction to help platforms and researchers find common ground for their collaborations. A degree of openness should be expected from companies when millions of users are at stake. At the same time, rigorous standards for handling such highly sensitive data should be expected from researchers.

The optimal outcome of collaboration between industry and academia is threefold. The platform is provided with an effective intervention, the research community has advanced the ecological validity of the intervention, and the users benefit from the collaboration without compromising their privacy. To achieve this optimal collaboration, we want to encourage the idea that a collaboration can involve multiple methods, carefully navigating the trade-offs of different approaches. For example, community-driven formats can counterbalance the problematic non-transparency of platform-internal projects by holding the platform accountable for full transparency of its research results. At the same time, internal project planning can use certain resources (e.g., direct modification of the platform environment) to more systematically experiment with the community involved.

### Words of caution
We would like to reiterate that any applicable intervention, although based on solid and replicable evidence, also has limitations and conditions. These limitations lie in the strength of the intervention, such as the duration of an intervention effect (i.e., no long-lasting effect), and in the generalizability of the intervention, such as cultural comparability (i.e., the intervention is differentially effective for different cultures). Practitioners should be aware of these limitations when applying interventions, and collaborations with researchers should systematically examine such intervention limitations as an opportunity to further our understanding of when and why an intervention is effective.

In this regard, we would also like to point out that existing digital interventions are mainly focused on reducing antisocial behavior (see, e.g., preliminary flagging, prebunking, and reminder of norms). Among the interventions presented in this paper, only a few aim to directly promote prosocial behavior (e.g., thank you button). This reflects a general imbalance: in our search for tested interventions, we also find few that aim to promote positive outcomes. To explore all possible approaches to creating a more prosocial digital space, future research should focus equally on preventing risks and promoting prosocial outcomes through digital interventions.

### Alternative intervention frameworks
We note that there are other novel approaches to categorizing digital interventions that are orthogonal to the practical framework presented here. Some of us[46] recently drew a distinction between behavioral and informative interventions, with the former targeting behavior change and the latter promoting users' understanding of the environment in which they act. The authors suggest that research has predominantly focused on behavioral interventions, while informative interventions are essential for campaigns that aim to promote users' competence and independence online. Our[46] categorization, however, is less focused on applicability and direct practicality, but rather on stimulating new avenues of intervention research. That is, their categorization had the primary goal of drawing attention (of scientists and practitioners) to a previously underexplored type of intervention (i.e., boosting) rather than inventing a classification framework for existing interventions.

Similarly, there is an ongoing scholarly debate about the use of recently conceptualized so-called boosts vs. the more well-known behavioral nudges to influence users' online behavior. Several of the digital interventions presented in our framework can be categorized as the former type of intervention, while others represent the latter type. For example, the tips and training on fact-checking and lateral reading[22,26] not only nudge behavior but also have additional long-term educational effects. In contrast, preliminary flagging[39,42] specifically prompts thought and behavior. We explicitly support the boosting type of intervention. Boosts invite a more detailed examination of intervention effects. Rather than simply emphasizing a preferred behavior, as nudges do, boosting interventions aim to foster people's internal competencies for effective self-directed action[47,48]. This interventional approach has several advantages. To name just two, boosts promote a higher degree of scalability because the central goal is to foster a self-sustaining competence in the user. Boosts are also less ethically objectionable: boosting relies on empowering users, which requires transparency, while nudging merely redirects users. This idea of distinguishing between boosts and nudges falls under the more general approach of categorizing interventions according to their (intended) outcome.

### Outlook
In charting a path for future research in this area, there are three clear categories of important interventions not included in this review that we hope to see in the future. First, there is a wide range of promising conceptual interventions based on theories tested in other (often offline) contexts that have yet to be tested or implemented, such as the bystander effect[49] and different persuasion techniques[50,51]. We hope that platform practitioners can work with researchers to co-create interventions that build on and translate a rich set of insights and knowledge into our online social spaces. Collaborations here should be particularly effective because of the combined expertise of the researchers with the particular intervention and the practitioners' knowledge of the new (i.e., digital) environment. We hasten to add that the rapid research circle in companies (compared to most academic institutions) makes strong and robust theorizing, indeed, more difficult. One solution to this is to, as much as possible, prepare theoretical considerations in form of reviews, frameworks, and guidelines for practitioners to make the inclusion of theoretical considerations more attractive in general by making it more efficient.

Second, and more critically, there are many interventions currently being used by platforms large and small that lack scientific evidence. Many platforms are releasing and testing creative and promising interventions, but the evidence of their effectiveness (if any) only appears in a company blog post or related press coverage. For example, in 2020, Nextdoor released a "Good Neighbor Pledge"[52] that builds on prior social science research on engagement and pledges to encourage platform users to help and respect one another. Similarly, Twitter, now known as X, tested an interactive intervention that prompted users who were in the process of retweeting an article to remind them to read the article

before posting it. In the press release announcing this feature, the company claims that "people who were shown the prompt opened articles 40% more often, and the overall proportion of people who opened articles before retweeting increased by 33 percent"[53]. These are just two examples of dozens, if not hundreds, of insightful experiments conducted by platforms that are not included in this review due to the lack of peer-reviewed evidence and the impossibility of fully reviewing the methods, data, and analyses. However, these interventions deserve the attention of the scientific community. Further, it also holds great potential for other practitioners and the organization itself to make the evidence behind platform-implemented interventions transparent beyond the walls of a single organization. Due to their direct implementation inside platforms, these interventions generally have high ecological validity and greater power to detect meaningful effect sizes. Findings from this research could therefore greatly benefit the body of knowledge by showing which pathways have been successful and which have not, and by stimulating new, more effective interventions. Importantly, as noted earlier regarding the current practitioner survey, designers, project managers, and other practitioners currently look to their peers and these corporate blog posts as an important source of evidence for decision-making. Thus, it is often the case that an intervention launched on one platform is quickly adopted by many of its peers. This is certainly a great thing if these interventions are very effective and their effects are generalizable across contexts, but without a scientific study of their efficacy, teams may be spending valuable and limited resources developing ineffective interventions just to keep up with their peers. Lastly, by making intervention evidence transparent, the organization itself can accelerate their development of effective interventions. Community and expert feedback from outside the organization can quickly point to what might go wrong with ineffective interventions and what could still be optimized in effective interventions.

Third, many of the interventions presented could be productively combined. In particular, one goal of collaboration between practitioners and researchers, which is also new for social platforms, could be to examine the interactive effects of different combinations of tested digital interventions. In particular, collaborative efforts would be needed to combine conceptual and methodological knowledge about interaction effects with knowledge about practical consequences and resources needed when combining certain interventions. To illustrate, the prebunking intervention could be well accompanied by tips for checking headlines, as both refer to a priori training of users' resistance to misinformation. While one increases automatic resistance to misinformation, the other adds the tools to consciously filter out such problematic information.

Lastly, the practical framework presented can be useful for practitioners to make sense of digital interventions in digital spaces other than social media. To name just a few, health care could be advanced by systematic research collaborations on digitized medication reminders as proactive, telemedicine consultations as interactive, and alerts of vital sign irregularities as reactive interventions[54,55]. In education, automated progress trackers could serve as useful proactive interventions, systematic virtual classrooms as interactive interventions, and adaptive learning systems as reactive interventions. Finally, intelligent public mobility provides predictive maintenance alerts as proactive, feedback and information assistants during travel as interactive, and systematized emergency response systems as reactive interventions.

Digital interventions promoting prosocial behavior are an essential part of navigating the complex environment of social media platforms online. Research on prosocial digital interventions is plentiful and many research directions are promising. While this growing body of research on digital interventions speaks to scholars' awareness of social problems in the digital space and their interest in reliable and effective moderators of online interaction, it remains unclear how readily these findings are used by practitioners. Part of the problem seems to be a missing common ground between the parties as the result of missing frameworks that make sense of the abundance of existing interventions and that can streamline collaborative efforts. Promoting common ground of understanding, above all, affords leveraging expertise from researchers, practitioners, and especially their collaborative thinking. Paving the way for these collaborations should be a central goal of our community.

## Data availability

The presented data of practitioners can be found on the following project page on OSF, Open Science Framework: https://osf.io/b3es6/?view_only=e5aa0dbeb16c4781986f340c38f89482.

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

## Author contributions

Contributed to practical framework idea: D.J.G., J.K., F.P., M.K., F.L.P. Contributed to the practical review: D.J.G., J.K., F.P., F.L.P. Contributed to repeated discussions of the contents: D.J.G., J.K., F.P., M.K., F.L.P. Drafted the first version of the article: D.J.G. Revised the article: D.J.G., J.K., F.P., M.K., F.L.P. Approved the submitted version for publication: D.J.G., J.K., F.P., M.K., F.L.P.

## Funding

## Competing interests

The authors declare the following competing interests: All authors are members of the registered non-profit organization *Prosocial Design Network*, which maintains the webpage https://www.prosocialdesign.org/.
