## [Peer Review File · Communications Psychology]

28th Mar 23

Dear Mr Grüning,

We have given the Perspective our careful consideration and find it of potential interest to the readers of Communications Psychology. However, we ask you to undertake a circumscribed set of revisions to make the work suitable for peer-review at the journal.

Presently, the work refers repeatedly to the project you and your co-authors lead, which is no doubt a highly valuable and well-presented resource. Nonetheless, to be suitable as a Perspective and to attract an engaged readership, your manuscript needs to work as a standalone reference for researchers and practitioners. This means that although it is of course acceptable to include one detailed pointer to the resource, the text must be distinct from the webpage and contain sufficient added value in the form of scholarly discussion; at the moment, the work likely to be misread as a precis of the website.

In particular, we would like you to add a critical, expert discussion of the papers/studies that tested each specific intervention. We would also ask you to limit reference to the prosocial design initiative to a single instance. I have provided feedback on your manuscript in the attachment to help guide your revisions. Your role at prosocial design should be acknowledged under conflicts of interest. We appreciate that the project is a registered non-profit; however, the nature portfolio asks for both, financial and nonfinancial competing interests to be declared.

We would therefore like to invite you to revise your manuscript to address these concerns.

We shall hope to receive your revised version as soon as you are able to complete the suggested revisions. If something similar is published in the interim we will have to consider the impact it has on the novelty of a revised manuscript.

If you anticipate a delay of more than four weeks, please let us know. We will be happy to consider your revision so long as nothing similar has been accepted for publication at Communications Psychology. Should your manuscript be substantially delayed without notifying us in advance and your article is eventually published, the received date may be that of the revised, not the original, version.

If you are not interested in submitting a suitably revised manuscript in the future please let me know immediately so we can close your file. If you have any questions, please contact me.

Please use the link below when you are prepared to resubmit.

[link redacted]

Thank you for your interest in Communications Psychology.

Best regards,
Jennifer Bellingtier

Jennifer Bellingtier, PhD

Senior Editor
Communications Psychology

5th Jul 23

Dear Mr. Grüning,

Thank you for your patience during the peer-review process. Your manuscript titled "A review-based framework of digital interventions for online prosocial behavior" has now been seen by 2 reviewers, and I include their comments at the end of this message.

The reviewers are in principle enthusiastic about the prospect of your work. However, they also mention a number of concerns.

Editorially, we think a significantly revised piece would be of high utility for our readers and suitable for publication in *Communications Psychology*. Before we make a final decision on publication, we would like to consider your response to these concerns in the form of a revised manuscript before we make a decision on publication.

Similarly to the recommendations made by Reviewer #2, and taking on board both of the referees' feedback, we believe the piece needs development to satisfy two aims. First, it should be readable and useful to "practitioners", such as platform developers. As you point out, there are significant barriers to non-scientists benefitting from scientific publications, and the presentation of the work is one of these. To achieve wide (applied) readership, the manuscript needs to provide a much clearer roadmap from your review of potential interventions to their application, with information relevant to the applied context. Second, the piece needs to dedicate more space to how scientists and practitioners can work together in a way beneficial to both parties (and potentially to the users of the platforms).

The required changes are substantive, not purely cosmetic. To aid you with that task, I have included a marked-up version of your manuscript. If you decide against undertaking these revisions, please let us know. Otherwise, we look forward to receiving a revision that takes all reviewer and editor comments into account.

EDITORIAL POLICIES AND FORMATTING

You will find a complete list of formatting requirements following this link:

<https://www.nature.com/documents/commsj-style-formatting-checklist-review-perspective.pdf>

Please use the checklist to prepare your manuscript for resubmission.

* **TRANSPARENT PEER REVIEW:** *Communications Psychology* uses a transparent peer review system. This means that we publish the editorial decision letters including Reviewers' comments to the authors and the author rebuttal letters online as a supplementary peer review file. We publish these records for all accepted manuscripts. However, on author request, confidential information and data can be removed from the published reviewer reports and rebuttal letters prior to publication. If your manuscript has been previously reviewed at another journal, those Reviewers' comments would not form part of the published peer review file.

If you have any questions about any of our policies or formatting, please don't hesitate to contact me.

Please use the following link to submit your revised manuscript and a point-by-point response to the referees' comments (which should be in a separate document to any cover letter):

[link redacted]

We hope to receive your revised paper within 12 weeks; please let us know if you aren't able to submit it within this time so that we can discuss how best to proceed. If we don't hear from you, and the revision process takes significantly longer, we may close your file.

We would appreciate it if you could keep us informed about an estimated timescale for resubmission, to facilitate our planning. Of course, if you are unable to estimate, we are happy to accommodate necessary extensions nevertheless.

Please do not hesitate to contact me if you have any questions or would like to discuss these revisions further. We look forward to seeing the revised manuscript and thank you for the opportunity to review your work.

Best regards,

Jennifer Bellingtier, PhD
Senior Editor
Communications Psychology

REVIEWERS' EXPERTISE:

Reviewer #1 positive interventions, digital interventions

Reviewer #2 media psychology, computer-mediated communication

REVIEWERS' COMMENTS:

Reviewer #1 (Remarks to the Author):

The authors present a rigorous and practically-minded review of digital interventions for online prosocial behavior. The manuscript is written in such a way that their work synthesizing this literature will provide great value to both researchers and industry product teams. Their use of both published and real-world examples adds further practical usefulness. As a fresh pair of eyes on this paper, I would also commend the authors on a number of thoughtful and thorough revisions that -- in the absence of context about what was asked for by reviewers -- undoubtedly improved the already strong manuscript.

My only suggestion would be to consider, at the very end, places where this work might come in handy outside of social media, where many of their examples came from. Healthcare, for example,

might be a very interesting context to apply this work to drive better health communication and subsequently better health literacy, and it may happen in the context of digital tools owned by health systems, or wearable/health app companies. This is really just a thought, but I would hate to see this paper miss the attention of those audiences, as it could meaningfully cross-pollinate there.

Reviewer #2 (Remarks to the Author):

I have now read “A review-based framework of digital interventions for online prosocial behavior”. The article fosters greater use of empirically validated interventions by developing a framework meant to spur collaborations between private industry and academic researchers. I believe this goal to be critically important for today’s media landscape and thus, the article has great potential to substantially contribute to both areas. I make the below suggestions in hopes of making their contribution more explicit and clearer to academics and practitioners.

Overarching Thoughts

Overall, I believe the manuscript’s contribution is being obscured by the lack of a clear target audience. From looking at the revisions, I’m guessing the original target audiences were both academic researchers and industry practitioners but then reviewers might have suggested only one. The revised sections suggest the chosen audience is academic researchers but, holistically, the manuscript still reads as a call to both academia and private industry for more collaborations. To me (and this may be in contrast to previous reviews) the call to both audiences makes more sense for this manuscript. I can understand focusing on academic researchers, given the barriers to reaching industry professionals as discussed by the authors, but I question the utility of providing researchers “practical deployment categories” ... “with a specific focus on the interventions’ practical applications”. That is, academic researchers are not equipped nor have the power to implement these interventions. Although the authors say this review can help researchers “address existing theoretical and empirical gaps”, I am hesitant to call this framework theoretical or theory-driven (see below). Thus, the main utility of this framework would mainly apply to the relatively small group of researchers with the resources to test the empirical and practical implications of such interventions on a large scale.

To me, the problem seems to be the framework is not particularly useful for researchers, but practitioners are also unlikely to use the framework if presented in this outlet. However, I think a potential solution is to embrace, what I feel is, the original spirit of the submission as a call to collaboration but additionally include recommendations for initiating and cultivating industry partnerships (something I’m guessing the authors have some experience with). In other words, the authors already detail the reasons why practitioners do not use science-back interventions and I think the article’s contribution could be to inform academics on how to overcome these barriers and directly connect with the industry, while also providing them an easily digestible and practical framework to use in those collaborations with practitioners. For example, the section “Alternative intervention frameworks” does not substantially contribute to the manuscript as those alternatives are more research focused, in contrast to the practical nature of the current framework. However, this section (and perhaps other parts of the general discussion) could focus on some recommendations for readers (depending on which area they reside in) that focus on building collaborations. Where should academics begin looking for potential collaborators (conferences, conventions, common university resources, etc)? Who should researchers even be looking for when

forming a team capable of creating such interventions? For practitioners who may come across the article, how can they identify relevant professors/researchers (online repositories/search engines, specific journals, etc.)? This still makes academic researchers the primary target audience but, as noted by the authors, I agree there are industry professionals who want to take advantage of academic research but simply don't know where to begin.

I know this deviates from the current content quite a bit but it might be worth thinking about for this outlet or others. Whatever the case may be, I also provide some specific thoughts regarding the manuscript structure that may be helpful down the line. However, the applicability of these suggestions will vary depending on the direction the authors take the manuscript.

- I believe the first two sections can present a stronger case and clearer roadmap for readers regarding the need for such a framework and its structure/contents. The current introductory paragraph establishes the need for this framework by saying “to the best of our knowledge, there exists no review with a practical system of categorization ...”. To me, this presents the framework's impetus as a simple taxonomy for previously unconnected research and, consequently, greatly undersells the larger goal of bridging research-backed interventions to real-world implementation.

Although other areas of the manuscript highlight this larger goal (which should be brought up into the introductory paragraph), I believe there is still more missing when it comes to setting up the specific focus on the chosen interventions. For example, the authors define digital interventions in the first paragraph as those that modify existing platforms, which naturally excludes any stand-alone interventions that may be used concurrently or in a complementary fashion. Without establishing the need and purpose of the framework earlier on, I was pretty confused by this restriction. As suggested in the “Digital Interventions of Interest” section, I would guess these interventions are the most practical for implementation by the platforms themselves. But I believe more space should be dedicated (particularly in the first few pages) to establishing the need for this framework and couching the framework within the larger goal of the manuscript. I believe these two things will help clarify the specific stipulations for certain interventions and why they are categorized in a certain manner from the beginning.

- Regarding the framework as theoretically based and theory-driven. Although the words theory/theoretical are used in the manuscript, I'm not convinced the framework can be called theoretical. The manuscript itself is inconsistent on what is driving the framework: the title says “review-based” and other parts say the framework is a “conceptually meaningful approach” or it presents “practical deployment categories”. I believe the latter two descriptions are more apt and the theory aspect seems post-hoc. Even if the manuscript were to embrace the cited theoretical foundation (“temporal starting points of interventions in nondigital environments”), I would still feel hesitant to call the framework theoretical. Unless there are some underlying theoretical processes in the cited work related to alcohol abuse (which should be elaborated on), the temporal aspect of the categorization seems more practical than theoretical. Perhaps the theoretical emphasis refers to the theories underlying the effectiveness of the interventions and not the chosen organization of the framework. If this is the case, then it should be made clear.

- Smaller note: The header “The need for scientifically tested interventions” and its section is confusing. I think the header is referring solely to the first paragraph in the section which may be more accurately labeled “The need for scientifically tested interventions in industry”. However, the second paragraph explains the whole purpose of the framework which either should be in its own

section, perhaps labeled by the framework's name or whatever moniker you want, or the original header should be relabeled to encompass both paragraphs.

I hope the authors find these thoughts helpful and I wish them the best of luck in their important work on prosocial behaviors.

John A. Velez, Ph.D.
Communication Science Unit
The Media School
Indiana University - Bloomington

Reviewer 1

- 1. The authors present a rigorous and practically-minded review of digital interventions for online prosocial behavior. The manuscript is written in such a way that their work synthesizing this literature will provide great value to both researchers and industry product teams. Their use of both published and real-world examples adds further practical usefulness. As a fresh pair of eyes on this paper, I would also commend the authors on a number of thoughtful and thorough revisions that -- in the absence of context about what was asked for by reviewers -- undoubtedly improved the already strong manuscript.*

We thank the reviewer very much for this encouraging comment. Especially the combination of the presented intervention cases being informed by published research and applied in the real world is an aspect that we made even more prominent in the revised manuscript.

- 2. My only suggestion would be to consider, at the very end, places where this work might come in handy outside of social media, where many of their examples came from. Healthcare, for example, might be a very interesting context to apply this work to drive better health communication and subsequently better health literacy, and it may happen in the context of digital tools owned by health systems, or wearable/health app companies. This is really just a thought, but I would hate to see this paper miss the attention of those audiences, as it could meaningfully cross-pollinate there.*

We fully agree that opening up the scope of application fields of digital interventions for prosociality should be a relevant goal in our scientific field and would be optimally integrateable in the present manuscript. Accordingly, we have now added this perspective in the last section of the revised manuscript, as the ending note of the outlook (p. 31) by applying the framework categorization on existing digital interventions in the diverse domains of healthcare, education, and public mobility.

Reviewer 2

1. *I have now read “A review-based framework of digital interventions for online prosocial behavior”. The article fosters greater use of empirically validated interventions by developing a framework meant to spur collaborations between private industry and academic researchers. I believe this goal to be critically important for today’s media landscape and thus, the article has great potential to substantially contribute to both areas. I make the below suggestions in hopes of making their contribution more explicit and clearer to academics and practitioners.*

We thank the reviewer very much for the evaluation that the manuscript has promising potential and the extensive and insightful comments provided. We have integrated all these suggestions in the revised manuscript.

2. *Overall, I believe the manuscript’s contribution is being obscured by the lack of a clear target audience. From looking at the revisions, I’m guessing the original target audiences were both academic researchers and industry practitioners but then reviewers might have suggested only one. The revised sections suggest the chosen audience is academic researchers but, holistically, the manuscript still reads as a call to both academia and private industry for more collaborations. To me (and this may be in contrast to previous reviews) the call to both audiences makes more sense for this manuscript. I can understand focusing on academic researchers, given the barriers to reaching industry professionals as discussed by the authors, but I question the utility of providing researchers “practical deployment categories” ... “with a specific focus on the interventions’ practical applications”. That is, academic researchers are not equipped nor have the power to implement these interventions. Although the authors say this review can help researchers “address existing theoretical and empirical gaps”, I am hesitant to call this framework theoretical or theory-driven (see below). Thus, the main utility of this framework would mainly apply to the relatively small group of researchers with the resources to test the empirical and practical implications of such interventions on a large scale.*

To me, the problem seems to be the framework is not particularly useful for researchers, but practitioners are also unlikely to use the framework if presented in this outlet. However, I think a potential solution is to embrace, what I feel is, the original spirit of the submission as a call to collaboration but additionally include recommendations for initiating and cultivating industry partnerships (something I’m guessing the authors have some experience with). In other words, the authors already detail the reasons why practitioners do not use

science-back interventions and I think the article's contribution could be to inform academics on how to overcome these barriers and directly connect with the industry, while also providing them an easily digestible and practical framework to use in those collaborations with practitioners. For example, the section "Alternative intervention frameworks" does not substantially contribute to the manuscript as those alternatives are more research focused, in contrast to the practical nature of the current framework. However, this section (and perhaps other parts of the general discussion) could focus on some recommendations for readers (depending on which area they reside in) that focus on building collaborations. Where should academics begin looking for potential collaborators (conferences, conventions, common university resources, etc)? Who should researchers even be looking for when forming a team capable of creating such interventions? For practitioners who may come across the article, how can they identify relevant professors/researchers (online repositories/search engines, specific journals, etc.)? This still makes academic researchers the primary target audience but, as noted by the authors, I agree there are industry professionals who want to take advantage of academic research but simply don't know where to begin.

I know this deviates from the current content quite a bit but it might be worth thinking about for this outlet or others.

We are very thankful about this in detail, more fundamental feedback. In accordance with the comment, we have now substantially shifted the overall focus of the revised manuscript toward practitioners. This is especially apparent in the sections after the introduction of all tested digital interventions. Here, we focused specifically on iterating how this framework becomes practical for practitioners (bringing the researcher efforts in these interventions into actual platform applications; p. 23 and 24). Additionally, we now (also in accordance with the editor's central feedback) provide an extensive section on how practitioners and researchers can collaborate in actuality, outlining existing and already used approaches with instructive comments on each format's advantages and challenges (p. 25 to 27).

- 3. I believe the first two sections can present a stronger case and clearer roadmap for readers regarding the need for such a framework and its structure/contents. The current introductory paragraph establishes the need for this framework by saying "to the best of our knowledge, there exists no review with a practical system of categorization ...". To me, this presents the framework's impetus as a simple taxonomy for previously unconnected research*

and, consequently, greatly undersells the larger goal of bridging research-backed interventions to real-world implementation.

Although other areas of the manuscript highlight this larger goal (which should be brought up into the introductory paragraph), I believe there is still more missing when it comes to setting up the specific focus on the chosen interventions. For example, the authors define digital interventions in the first paragraph as those that modify existing platforms, which naturally excludes any stand-alone interventions that may be used concurrently or in a complementary fashion. Without establishing the need and purpose of the framework earlier on, I was pretty confused by this restriction. As suggested in the “Digital Interventions of Interest” section, I would guess these interventions are the most practical for implementation by the platforms themselves. But I believe more space should be dedicated (particularly in the first few pages) to establishing the need for this framework and couching the framework within the larger goal of the manuscript. I believe these two things will help clarify the specific stipulations for certain interventions and why they are categorized in a certain manner from the beginning.

We fully agree that especially the first section undersells the whole idea of the framework to establish not just an overview but interactive forces between practitioners and researchers to result in a more collaborative future of testing and implementing digital interventions for prosocialty. In the revised manuscript, we now adapted this to be substantially more explicit as our main goal of the paper (p. 4 and 5). Additionally, we make the fact very prominent that the framework presented here is specifically focused on interventions on platforms due to this format being the most prominently discussed and researched one that exists (p. 4).

- 4. Regarding the framework as theoretically based and theory-driven. Although the words theory/theoretical are used in the manuscript, I'm not convinced the framework can be called theoretical. The manuscript itself is inconsistent on what is driving the framework: the title says “review-based” and other parts say the framework is a “conceptually meaningful approach” or it presents “practical deployment categories”. I believe the latter two descriptions are more apt and the theory aspect seems post-hoc. Even if the manuscript were to embrace the cited theoretical foundation (“temporal starting points of interventions in nondigital environments”), I would still feel hesitant to call the framework theoretical. Unless there are some underlying theoretical processes in the cited work related to alcohol abuse (which should be elaborated on), the temporal aspect of the categorization seems more practical than theoretical. Perhaps the theoretical emphasis refers to the theories*

underlying the effectiveness of the interventions and not the chosen organization of the framework. If this is the case, then it should be made clear.

We fully agree with this evaluation and have, accordingly, changed the reference to our framework as, first and foremost, practical. This also is the optimal communication of our focus in the revised manuscript, given that we are substantially changing the perspective toward practitioners.

5. *The header “The need for scientifically tested interventions” and its section is confusing. I think the header is referring solely to the first paragraph in the section which may be more accurately labeled “The need for scientifically tested interventions in industry”. However, the second paragraph explains the whole purpose of the framework which either should be in its own section, perhaps labeled by the framework’s name or whatever moniker you want, or the original header should be relabeled to encompass both paragraphs.*

We thank the reviewer for this relevant note. Accordingly, we have now made the second paragraph the first paragraph of the larger chapter on the framework. This paragraph is now the introduction that leads to the subchapters of the framework, namely, which interventions are included (i.e., “Digital interventions of interest”) and how the interventions are categorized (i.e., “Categorizing interventions by deployment time”).

17th Nov 23

Dear Mr Grüning,

Your Review Article titled "A framework of digital interventions for online prosocial behavior" has now been seen by 2 referees, whose comments appear below. In the light of their advice I am delighted to say that we are happy, in principle, to publish it in Communications Psychology under a Creative Commons 'CC BY' open access license.

We will not send your revised paper for further review if, in the editors' judgement, the referees' comments on the present version have been addressed. If the revised paper is in Communications Psychology format, in accessible style and of appropriate length, we shall accept it for publication immediately. I have attached an edited version of your manuscript, and ask you to attend to each comment in detail.

EDITORIAL REQUESTS:

* Please review the changes in the attached copy of your manuscript, which has been edited for style, and address the comments and queries I have added. If using Word, please use the 'track changes' feature to make the process of accepting your manuscript more efficient.

* Communications Psychology uses a transparent peer review system. On author request, confidential information and data can be removed from the published reviewer reports and rebuttal letters prior to publication. If you are concerned about the release of confidential data, please let us know specifically what information you would like to have removed. Please note that we cannot incorporate redactions for any other reasons.

*If you have not done so already, please alert me to any related manuscripts from your group that are under consideration or in press at other journals, or are being written up for submission to other journals (see www.nature.com/authors/editorial_policies/duplicate.html for details).

FORMATTING GUIDELINES:

I have attached a Microsoft Word document, our Editorial Request Table, that specifies exactly how your manuscript must be formatted to be accepted for publication. In addition to revising your paper, you will need to complete the Editorial Request Table (which serves as a checklist) and upload it with your revision. I also highlight a few issues that are of particular importance below:

** Title

Titles should be descriptive of the main message your manuscript conveys and should not exceed 90 characters (including spaces). Although the choice of title is largely yours, may I suggest the following:

A framework for promoting online prosocial behavior via digital interventions

** Length

The ideal length for Review Articles and Perspectives in Communications Psychology is 5,000 words.

Your Perspective is beyond that limit, which is acceptable in the present case, but please do not make it any longer.

* References

References appear as superscript Arabic numerals, in order of mention. The reference list mentions references in the numerical order in which they are mentioned in the main text. If a reference is cited more than once, the same number is used throughout the text and the reference receives a single entry in the reference list.

We ask that you select the most significant 5–10% of references in your list for highlighting, and add a single sentence in bold after each of these references to describe the main result and its significance.

Only papers that have been published or accepted by a named publication should be in the reference list (preprints and citations of datasets are also permitted). Unpublished/Submitted research should not be included in the reference list; it should only be mentioned briefly and parenthetically in the main text. Note that no major arguments should rely on unpublished research.

Published conference abstracts and URLs for web sites should be cited parenthetically in the text, not in the reference list.

Footnotes are not used.

SUBMISSION INFORMATION:

* Your paper will be accompanied by an editor's summary, of between 250-300 characters, when it is published on our homepage. Could you please approve the draft summary below or provide us with a suitably edited version.

Grüning and colleagues review digital interventions for promoting prosocial online behavior and provide insights and tips for collaborations between academic researchers and practitioners.

In order to accept your paper, we require the following:

* A cover letter describing your response to our editorial requests.

* The final version of your text as a Word or TeX/LaTeX file, with any tables prepared using the Table menu in Word or the table environment in TeX/LaTeX and using the 'track changes' feature in Word.

* Production-quality versions of all figures, supplied as separate files. Photographic images should be 300 dpi in RGB format (.jpg, TIFF or native Photoshop format) and any labels/scale bars included in a separate layer from the image. Line art, graphs and schemes should be vector format (.ai, .eps, .pdf); Adobe Illustrator files are preferred and will minimize production time. Any chemical structures or schemes contained within figures should additionally be supplied as separate Chemdraw (.cdx) files.

At acceptance, the corresponding author will be required to complete an Open Access Licence to Publish on behalf of all authors, declare that all required third party permissions have been obtained.

Please note that your paper cannot be sent for typesetting to our production team until we have received this information; **therefore, please ensure that you have this ready when submitting the final version of your manuscript.**

ORCID

Communications Psychology is committed to improving transparency in authorship. As part of our efforts in this direction, we are now requesting that all authors identified as 'corresponding author' create and link their Open Researcher and Contributor Identifier (ORCID) with their account on the Manuscript Tracking System (MTS) prior to acceptance. ORCID helps the scientific community achieve unambiguous attribution of all scholarly contributions. For more information please visit <http://www.springernature.com/orcid>

For all corresponding authors listed on the manuscript, please follow the instructions in the link below to link your ORCID to your account on our MTS before submitting the final version of the manuscript. If you do not yet have an ORCID you will be able to create one in minutes.

IMPORTANT: All authors identified as 'corresponding author' on the manuscript must follow these instructions. Non-corresponding authors do not have to link their ORCIDs but are encouraged to do so. Please note that it will not be possible to add/modify ORCIDs at proof. Thus, if they wish to have their ORCID added to the paper they must also follow the above procedure prior to acceptance.

To support ORCID's aims, we only allow a single ORCID identifier to be attached to one account. If you have any issues attaching an ORCID identifier to your MTS account, please contact the Platform Support Helpdesk.

[link redacted]

We hope to hear from you within two weeks; please let us know if the process may take longer.

Best regards,

Jennifer Bellingtier

Jennifer Bellingtier, PhD

Senior Editor
Communications Psychology

REVIEWERS' EXPERTISE:

Reviewer #1 positive interventions, digital interventions

Reviewer #2 media psychology, computer-mediated communication

REVIEWERS' COMMENTS:

Reviewer #1 (Remarks to the Author):

Comments adequately addressed.

Reviewer #2 (Remarks to the Author):

I appreciate the time and effort the authors have put into the revisions. I believe the manuscript provides a clear and substantial contribution to potential practitioners. I have one more major consideration for the authors and it revolves around framing the interventions as reducing antisocial behaviors instead of promoting prosocial behaviors.

On page 9, it says "Notably, the majority of digital interventions do not directly promote prosocial behavior, but rather attempt to reduce antisocial behavior...". This made me wonder why the title/headers all seem to suggest the opposite. I can't see a reason why framing the whole paper as promoting prosocial behavior would be preferred over reducing antisocial behavior. Particularly because this sentence makes the latter seem more accurate. I understand the ultimate goal is to promote prosocial behavior but it doesn't seem to be the actual goal of the interventions nor reflect the processes they utilize, which may confuse practitioners especially.

Now it's probably safe to assume all online digital contexts have antisocial behaviors and thus by reducing them you are always promoting prosocial behaviors, in a sense. However, I also wouldn't define prosocial behaviors as the absence of antisocial ones. That is, by reducing antisocial behaviors you are more accurately increasing the opportunity for prosocial behaviors to flourish, but not actually promoting them. This doesn't sound as impactful as saying these interventions are promoting prosocial behaviors but my point is that reducing antisocial behaviors is just as important (maybe more important given its prevalence) than promoting prosocial ones.

I also think the reduction of antisocial behaviors will be more appealing to practitioners. Sadly, unless a company's brand or product depends on prosocial behaviors, companies are more likely to search for interventions that reduce antisocial behaviors that obstruct the use of their platform. That is, I believe companies will be more driven to search for interventions when antisocial behaviors are affecting their bottom line, versus a desire to promote prosocial behaviors. Therefore, I think this article is more likely to gain the attention of practitioners if it more accurately reflects the goal of reducing antisocial behaviors. In terms of structure, this would also allow the authors to frame the Thank You Button intervention as a subsequent/supplemental tool to promote prosocial behaviors after successfully reducing antisocial ones with the other interventions. As of right now, the Thank You Button seems out of place.

Smaller Considerations:

Rereading the data from the survey reminded me that companies have whole departments dedicated to UX research/testing and many of these departments consist of academics/Ph.D.s. Indeed, I know a growing list of graduate students entering these departments or are interested in doing so. I imagine the presence of these departments makes designers less reliant on academic research given their main purpose is to conduct proprietary research specific to the company's products. This is now discussed in the revisions under the section "Outlook" which is great. I'm not sure how/when/where but it seems like this article should, at places, specifically target these departments/researchers. I should note that in discussions with past students working in the industry, they say that deductive theorizing based on past academic research is very impractical given the extremely rapid timeframes for studies/research (i.e., multiple studies conducted in a single week). But perhaps, this article could provide such researchers with a more practical use of academic research while also potentially convincing other designers/practitioners that such work is valuable to their company. I don't have specific revisions in mind but just thought it might be worth considering before publishing.

I uploaded the article with comments for the authors. Some are more important than others but I think a major one is the absence of a conclusion paragraph/section.

Thanks,
John A. Velez
Communication Science Unit
The Media School
Indiana University - Bloomington

Reviewer 1

- 1. Comments adequately addressed.*

We are delighted that our revisions seem to have addressed all posed ideas and comments.

Reviewer 2

1. I appreciate the time and effort the authors have put into the revisions. I believe the manuscript provides a clear and substantial contribution to potential practitioners. I have one more major consideration for the authors and it revolves around framing the interventions as reducing antisocial behaviors instead of promoting prosocial behaviors.

On page 9, it says "Notably, the majority of digital interventions do not directly promote prosocial behavior, but rather attempt to reduce antisocial behavior...". This made me wonder why the title/headers all seem to suggest the opposite. I can't see a reason why framing the whole paper as promoting prosocial behavior would be preferred over reducing antisocial behavior. Particularly because this sentence makes the latter seem more accurate. I understand the ultimate goal is to promote prosocial behavior but it doesn't seem to be the actual goal of the interventions nor reflect the processes they utilize, which may confuse practitioners especially.

Now it's probably safe to assume all online digital contexts have antisocial behaviors and thus by reducing them you are always promoting prosocial behaviors, in a sense. However, I also wouldn't define prosocial behaviors as the absence of antisocial ones. That is, by reducing antisocial behaviors you are more accurately increasing the opportunity for prosocial behaviors to flourish, but not actually promoting them. This doesn't sound as impactful as saying these interventions are promoting prosocial behaviors but my point is that reducing antisocial behaviors is just as important (maybe more important given its prevalence) than promoting prosocial ones.

I also think the reduction of antisocial behaviors will be more appealing to practitioners. Sadly, unless a company's brand or product depends on prosocial behaviors, companies are more likely to search for interventions that reduce antisocial behaviors that obstruct the use of their platform. That is, I believe companies will be more driven to search for interventions when antisocial behaviors are affecting their bottom line, versus a desire to promote prosocial behaviors. Therefore, I think this article is more likely to gain the attention of practitioners if it more accurately reflects the goal of reducing antisocial behaviors. In terms of structure, this would also allow the authors to frame the Thank You Button intervention as a subsequent/supplemental tool to promote prosocial behaviors after successfully reducing antisocial ones with the other interventions. As of right now, the Thank You Button seems out of place.

We thank the reviewer very much for this extensive comment. We discussed this alternative framing intensively and decided against it, arguing that while many existing and tested digital interventions are closely linked to reducing antisocial behavior, the general notion under which such research operates and also the overarching ideas and plea of the present paper is the promotion of prosocial behavior by digital interventions. Specifically, although much of the intervention focus has not been on prosocial interventions, the present paper has the central goal to review existing evidence and present a framework that promotes intervention effects on prosocial behavior.

- 2. Rereading the data from the survey reminded me that companies have whole departments dedicated to UX research/testing and many of these departments consist of academics/Ph.D.s. Indeed, I know a growing list of graduate students entering these departments or are interested in doing so. I imagine the presence of these departments makes designers less reliant on academic research given their main purpose is to conduct proprietary research specific to the company's products. This is now discussed in the revisions under the section "Outlook" which is great. I'm not sure how/when/where but it seems like this article should, at places, specifically target these departments/researchers. I should note that in discussions with past students working in the industry, they say that deductive theorizing based on past academic research is very impractical given the extremely rapid timeframes for studies/research (i.e., multiple studies conducted in a single week). But perhaps, this article could provide such researchers with a more practical use of academic research while also potentially convincing other designers/practitioners that such work is valuable to their company. I don't have specific revisions in mind but just thought it might be worth considering before publishing.*

We thank the reviewer for this comment and have added these aspects more prominently in the Outlook section now (p. 28). We especially prompted the fact that the rapid research circle in companies (compared to most academic institutions) makes strong and robust theorizing, indeed, more difficult. A solution to this is to, as much as possible, prepare theoretical considerations in form of reviews, frameworks, and guidelines for these practitioners to make the inclusion of theoretical considerations more attractive in general (by making it more efficient). The present paper, in fact, follows exactly this approach.